# Role of Dietary Fibre in Managing Periodontal Diseases—A Systematic Review and Meta-Analysis of Human Intervention Studies

**DOI:** 10.3390/nu15184034

**Published:** 2023-09-18

**Authors:** Hasinthi Swarnamali, Nidhi Medara, Aditi Chopra, Axel Spahr, Thilini N. Jayasinghe

**Affiliations:** 1The Liggins Institute, The University of Auckland, 85 Park Road, Grafton, Auckland 1023, New Zealand; 2Sydney Dental School, Faculty of Medicine and Health, The University of Sydney, Camperdown, NSW 2006, Australia; nidhi.medara@sydney.edu.au (N.M.); axel.spahr@sydney.edu.au (A.S.); thilini.jayasinghe@sydney.edu.au (T.N.J.); 3Department of Periodontology, Manipal College of Dental Sciences, Manipal, Manipal Academy of Higher Education, Manipal 576104, India; aditi.chopra@manipal.edu; 4The Charles Perkins Centre, The University of Sydney, Camperdown, NSW 2006, Australia

**Keywords:** dietary fibre, periodontal diseases, inflammation, periodontitis, gingivitis, humans

## Abstract

Background: Periodontitis is a chronic multifactorial inflammatory disease, that leads to tooth loss and is associated with other systemic diseases. The role of dietary fibre in the prevention and management of periodontal diseases is not well understood. The objective of this systematic review and meta-analysis was to assess how an intake of dietary fibre affects periodontal diseases in humans and any concomitant effects on systemic inflammation. Methodology: Human interventional studies investigating the effects of oral fibre intake on various clinical parameters of periodontal diseases were included. Search strategy with MeSH and free-text search terms was performed on the following database: CINAHL Complete, EMBASE, MEDLINE, SciVerse Scopus^®^, and Web of Science Core Collection on 21 October 2021 and updated on 19 February 2023 to identify relevant studies. Articles were filtered using the Covidence© web-based platform software. Data were pooled using random effects meta-analysis. Results: From all databases, a total of 19,971 studies were obtained. Upon title and abstract screening, 101 studies were included for full-text screening. Upon full-text screening, six studies were included for analysis. Of these, five were randomised controlled trials, and one was a sequential feeding trial involving fibre-rich daily diet for a 4–8 weeks period. Fibre-rich dietary intervention significantly reduced Clinical Attachment Loss/Level by 0.48 mm/tooth (95% CI, −0.63 to −0.33, *p* < 0.001), Bleeding On Probing by 27.57% sites/tooth (95% CI −50.40 to −4.74, *p* = 0.02), Periodontal Inflamed Surface Area by 173.88 mm^2^ (95% CI −288.06 to −59.69, *p* = 0.003), Plaque Index by 0.02 (95% CI −0.04 to −0.00, *p* = 0.04), and Gingival Index by 0.41 (95% −0.67 to −0.16, *p*= 0.002). A non-significant reduction was observed for Probing Depth (−0.17 mm/tooth; 95% CI, −0.37 to 0.02, *p* = 0.09). Conclusions: Fibre-rich dietary interventions are associated with a reduction of clinical and particularly inflammatory markers of periodontal diseases. This shows a promising effect of dietary fibre as an intervention for inflammatory conditions like periodontal diseases.

## 1. Introduction

Periodontal diseases are a significant public health concern worldwide as they are widespread and impede one’s quality of life and well-being [1]. They are one of the top six chronic non-communicable diseases globally [2]. Over one billion cases of severe periodontal diseases are estimated worldwide, affecting about 19% of the adult population [3]. Periodontitis, the most advanced form of periodontal disease, is a chronic multifactorial inflammatory disease associated with dysbiotic plaque biofilms and characterized by a progressive destruction of the tooth-supportive tissues, including gingiva and bone. It is one of the primary reasons for tooth loss among adults [4]. Periodontitis or its associated tissue destruction is irreversible. Its precursor or early stage, however, gingivitis, is characterised by a restricted inflammation of the gingiva only and is, subject to proper treatment, completely reversible without any remaining damage or tissue loss.

As teeth are the only natural non-shedding surfaces in a dynamic moist environment exposed to both external and internal factors, it is an optimum condition for microbial biofilm formation, which is one of the major aetiological factors for periodontal diseases [5]. This interaction of oral microorganisms and periodontal tissues leads to inflammation and gradual destruction of the periodontal tissues, including alveolar bone. This ultimately results in tooth loss [6].

The host-microbe interactions of periodontal diseases can be modified by various behavioural, environmental, and genetic risk factors as well as epigenetic effects. Nutrition is one of the behavioural and modifiable risk factors for periodontal diseases, and changes could be implemented on a population basis by patients themselves [7]. The composition of the oral microbiome is in a constant state of flux, influenced by factors such as diet, lifestyle habits (such as smoking), oral hygiene practices, medication use, hormonal conditions, the presence of dental prosthesis (like dentures and implants), and systemic diseases [8]. It is important to note that the microbial composition around natural teeth is different from that in the peri-implant environment. These distinctions primarily involve variations in bacterial populations within the classes Bacteroidia, Spirochaetes, Synergistia, Clostridia, and Deltaproteobacteria [9]. Therefore, dietary modifications can play a crucial role in influencing the microbiota and may hold significant potential for managing both periodontal and peri-implant diseases.

Studies have found that a balanced diet with proper nutrition is important for maintaining gingival and periodontal health [10,11]. According to the 2017 classification, diet is one of the important systemic factors that can modify the immune-inflammatory response [12]. Dietary components like fiber and essential vitamins are widely acknowledged for their role in promoting overall health [13]. Compared to our ancestors’ dietary habits, modern dietary practices have seen a marked decrease in fibre intake. This is linked to the introduction of low-fibre Western diets and is related to a higher prevalence of various health conditions, including inflammatory bowel disease, obesity, Type II diabetes mellitus, metabolic syndrome, and oral diseases [14,15,16]. To address this, the American Heart Foundation recommends a daily fibre intake of 30 g for men and 25 g for women [17]. Dietary fibre affects the host by specifically stimulating the growth and/or activity of one or a small number of bacteria in the colon, which improves host health [18] and by reducing body weight or limiting obesity-associated systemic inflammation [19,20]. In periodontal diseases, fibre decreases systemic and local inflammation in periodontal tissues and improves periodontal disease indices in humans [21,22,23].

The burden of periodontal diseases is substantial, and enhancing our understanding of the relation between dietary fibre intake and periodontal diseases could contribute to dietary recommendations impacting public health. Dietary interventions for managing periodontal diseases could become one of the promising customised and minimally invasive procedures to modulate oral microbiota and control periodontal inflammation [12]. However, it is important to note that most evidence often used to back oral health benefits of dietary fibre is mainly based on observational studies with inferior methodology. While the potential beneficial effect of dietary fibre on oral health is intriguing, they have not been systematically explained in the context of periodontal diseases. Additionally, most of the beneficial properties of dietary fibre have been derived from animal studies or small isolated clinical trials. To the best of our knowledge, no systematic reviews have comprehensively evaluated the effect of dietary fibre intake on periodontal diseases. Therefore, we aimed to systematically evaluate the effects of dietary fibre intake on the clinical parameters of adult patients with periodontal diseases in interventional studies. We also aimed to assess whether dietary fibre intake can modulate both local and systemic inflammatory markers.

## 2. Methodology

The study protocol was registered with the National Institute of Health Research PROSPERO, International Prospective Register of Systematic Reviews (https://www.crd.york.ac.uk/PROSPERO/ [accessed on 13 November 2021], registration number CRD42021284997), and the systematic review was conducted according to the Preferred Reporting Items for Systematic Review and Meta-Analysis (PRISMA) guidelines [24].

### 2.1. Inclusion and Exclusion Criteria

**Population (P):** Studies with participants having the following periodontal diseases according to the American Academy of Periodontology (AAP) 1999 and 2017 classifications were included: gingivitis (dental biofilm-induced), necrotizing periodontal disease, periodontitis as a manifestation of systemic diseases and periodontitis. Studies where participants reported any habits such as smoking, tobacco chewing, use of the smokeless or chewing form of tobacco, areca nut or supari were excluded. Studies with subjects undergoing chemotherapy, or radiation therapy and pregnant or lactating females were also excluded.**Intervention (I)/Exposure (E):** The intervention was dietary fibre (roughage or bulk), defined as parts of plant-derived food that the gut cannot absorb. Studies with any dose or type of dietary fibre in any form (part of the diet or supplementary as liquid or capsule) administered via the oral cavity were included. Studies where dietary fibre was used as an adjunct to non-surgical treatment (scaling and root planning ([SRP])) were also considered.**Comparator/Control (C):** Studies where the comparator was as follows: SRP alone, SRP + low fibre, low fibre alone, SRP + other soluble fibre, soluble fibre alone, or no fibre.**Outcomes (O):** The primary outcomes were changes in clinical Probing Depth/pocket Probing Depth (PD) and Clinical Attachment Level/loss (CAL). Other clinical parameters, Bleeding on Probing (BOP), Gingival Index (GI), sulcus bleeding index, Plaque Index (PI), gingival recession, Periodontal Inflamed Surface Area index (PISA), oral microbial outcomes or inflammatory biomarkers and metabolites in blood, Gingival Crevicular Fluid (GCF), saliva or oral tissues, were included as secondary outcomes. If the studies reported any other parameters, these were also included.**Types of study:** Randomised controlled trials (RCTs), quasi-randomised trials, and pre-clinical trials (e.g., cross-over studies, sequential feeding trials ([SFT]), parallel studies) evaluating the efficacy of fibre on periodontal diseases for any study duration were included following the PICO strategy. Observational studies (e.g., cohort, case-control, and cross-sectional studies), in-vitro models, case reports, case series, letters to the editor, reviews, unpublished data, and secondary literature on the effect of dietary fibre and periodontal diseases were excluded.

### 2.2. Information Sources and Search Strategies

The following database systems were used to perform the initial comprehensive electronic search: MEDLINE via Ovid, SciVerse Scopus^®^ (Elsevier Properties, Philadelphia, PA, USA), Web of Science [v.5.4] (Thomson Reuters, Eagan, MN, USA) Core Collection databases, CINAHL^®^ Complete, and EMBASE via Ovid SP (Elsevier Properties, Amsterdam, The Netherlands). Relevant studies were identified in the databases above by combining medical subject headings (MeSH) and free-text terms related to periodontal diseases and dietary fibre (Appendix A). Preliminary searches were performed from their oldest record to 21 October 2021. A comprehensive search strategy with MeSH and free text search terms was performed to update an original search before the study selection process was piloted (7 March 2022). The search was not limited by language or publication period. Reference lists from included studies, review articles, and online sources were searched manually to collect additional articles related to our PICO. Before the final analysis, searches were conducted again (1 August 2023), and any new studies were retrieved for inclusion.

### 2.3. Study Selection

The articles identified from the search strategy were imported to Covidence^©^ systematic review software- Version 2.0 (Veritas Health Innovation, Melbourne, Australia), and duplicates were eliminated. Title and abstract screening and full-text review based on the eligibility criteria were performed independently by two reviewers (HS and TNJ). Any disagreements were resolved by discussion and consensus with other reviewers (AC and NM).

### 2.4. Data Extraction

One reviewer (HS) extracted data from the included studies, and other reviewers verified the content’s accuracy (AC, NM and TNJ). The following information was collected:(1)General study details (first author, country, year of publication);(2)Study design and intervention period;(3)Diagnostic criteria defining the periodontal diseases in the given study;(4)Participant characteristics (health status of the participants, sample size of intervention vs. control groups, male and female distribution of intervention vs. control groups, age range or mean age of intervention vs. control groups);(5)Characteristics of intervention and comparator (type of fibre, form of intake [as part of the diet or oral supplement], dosage [grams per day]);(6)Evaluated outcomes:Primary outcomes: PD (mm/tooth), CAL (mm/tooth)Secondary outcomesOther metabolic parameters;(7)Effect of oral fibre intake compared to the comparison group (*p* values were presented);(8)Comparisons of outcome measures.

### 2.5. Risk of Bias and Methodological Quality Assessment

The Cochrane Handbook for Systematic Reviews of Interventions was used to appraise the risk for each randomised controlled trial [25].

The methodological quality of the included clinical trials was also evaluated according to the modified Jadad scale [26,27]. Each study was given a score ranging from 0 (poor quality) to 5 (good quality) based on the following parameters:(1)Was the study referred to as being random?(2)Was the study referred to as being double-blind/single-blind?(3)Was there a description of dropouts and withdrawals?(4)Was the paper’s described randomization method appropriate or not?(5)Was the described and appropriate blinding technique used?

The studies scored one point if the answer to questions 1 and 3 was “yes” and zero points if the answer was “no”. The second question was scored 1 or 0.5 points if the blinding process was double and single, respectively. If the method of randomisation was described but not appropriate, or if the method of blinding was described but not appropriate, questions 4 and 5 would receive a −1 score.

### 2.6. Quantitative Data Synthesis

Review Manager (RevMan) software was used for outcome meta-analyses (version 5.3 for Windows; The Nordic Cochrane Centre, Copenhagen, Denmark). The overall impact on periodontal disease parameters was expressed using the mean difference with 95% Confidence Interval (CI). All other habitual or standard diets were pitted against fibre-rich dietary approaches (interventions) as a comparison group. A statistically significant *p*-value of less than 0.05 was set for all analyses. A random-effects method was used for the meta-analysis due to variations in the design, population, types of control diets, and quantity of dietary fibre consumed across trials.

## 3. Results

### 3.1. Results of the Search

The database search using the search strategy retrieved 19,971 articles. After removing duplicated studies (*n* = 9117), 10,856 remained. Of these, 101 articles were included for full-text review. Ninety-five articles were excluded (Appendix A), and six studies were included in this review based on the eligibility criteria (Figure 1).

### 3.2. Study Characteristics

Three clinical trials from Germany [28,29,30], one from Japan [31], one from Switzerland [32] and one from India [33] evaluated the effect of dietary fibre intake on periodontal diseases. Of the six studies, five studies were RCTs ([29,30,31,32,33]), while the other was a SFT [31]. All studies involved adult population (age > 18 years). The mean age of the study groups ranged from 27.2 ± 4.7 to 45.0 ± 6.5 years, while the sample size ranged from 15 to 54. All six studies included participants from both genders and the ratio of male to female participants is given in Table 1. While Bartha et al. (2022), Rajaram et al. (2021), Woelber et al. (2016), Woelber et al. (2019), and Tennert et al. (2020) used participants with gingivitis (with the mentioned diagnostic criteria in Table 1) ([28,29,30,32,33]), Kondo et al. (2014) used high-risk individuals for periodontal diseases with body mass index (BMI) of at least 25.0 kg/m^2^ or a plasma glucose level at 2 h after a 75-g oral glucose tolerance test of at least 120 mg/dL [31].

Table 1 summarises the six clinical trials included and the outcomes of the clinical variables. Table 2 shows inter- and intra-*p* values between and within the intervention and control groups for periodontal outcomes.

### 3.3. Intervention Characteristics

All six interventions incorporated fibre into the diet without giving it as a supplement. Bartha et al. (2022) study used a Mediterranean diet with fibre content ranging from 22.10–26.06 g for the intervention group compared to 15.72–16.22 g fibre content for the control group [30]. The intervention lasted six weeks after the 2-week equilibration phase.

The SFT by Kondo et al. (2014) described the intervention period as a run-in period (duration was not specified), an 8-week test-meal period (which is considered a high-fibre intervention period), and a 24-week follow-up period [31]. The test-meal period used high-fibre content of 30.3 g of fibre daily in the diet. In comparison, fibre content during the run-in period and 24-week follow-up period remained similar at 11.9 g and 10.9 g, respectively. The fibre amount in the intervention (test-meal) period was three-fold higher than in the control (run-in) or follow-up period.

The Rajaram et al. (2021), Woelber et al. (2016), and Woelber et al. (2019) studies used similar diet recommendations [28,29,33]. Although they did not specify the exact dosage of fibre, dietary recommendations for the high-fibre or intervention group were based on literature concerning general inflammation [34,35] and gingival or periodontal inflammation [36,37].

In the Rajaram et al. (2021) study, the high-fibre intervention group consisted of a diet with low carbohydrates (<130 g/d intake with minimal intake of fructose, flour-containing food, rice, sweetened meals, beverages, and potatoes) and recommended amount of omega-3/6 fatty acids from recommended seeds and oils, ascorbic acid from citrus fruits, antioxidants from green tea/coffee/fruits, and fibre from vegetables and fruits while the control group followed a habitual diet without any dietary recommendation for four weeks [33]. In the intervention group, the degree of compliance to the dietary recommendation of fibre increased to 0.94 ± 0.07 at the end of the study period compared to 0.26 ± 0.20 at baseline (0 = no compliance, 1 = 100% consumption as recommended or complete compliance). The control group’s fibre compliance remained unchanged at 0.37 ± 0.08 at baseline and 0.38 ± 0.12 at the end [33].

In the Woelber et al. (2016) [28] study, the intervention group used the same diet protocol as the Rajaram et al. (2021) [33] study in addition to vitamin D from sun, supplementation, or avocado, while the control group followed a habitual diet without a dietary recommendation for eight weeks [28]. The degree of compliance to the dietary recommendation in the intervention group increased from 0.40, 0.38, 0.90, 0.88, 0.94, and 0.93 from weeks 1 to 8, whereas the compliance was 0.50, 0.64, 0.44, 0.48, 0.48, and 0.48 in the control group (degree of compliance after analyzing food diaries; 0 = no compliance, 1 = 100% consumption as recommended) [28].

The Woelber et al. (2019) [29] study used the same diet protocol as the Rajaram et al. (2021) [33] study in addition to a reduction of industrial animal proteins, vitamin D from sun or supplements, and fibre from vegetables, fruits, legumes, and bran, and nitrate-containing plants such as spinach, beetroot, or rocket, while the control group followed the habitual diet without any dietary recommendation for 4 weeks [29]. From baseline to week 4, fibre intake increased from 18.7 ± 17.05 to 39.06 ± 14.94 g in the intervention group but remained similar in the control group at 17.54 ± 7.63 g to 16.62 ± 8.65 g, respectively [29].

The intervention group in the Tennert et al. (2020) [32] study used oral health-optimised diet, including fibre-rich foods from fruits and vegetables based on recommendations for gingival and periodontal inflammation [34,38]. The intervention group used the same diet protocol as the Rajaram et al. (2021) [33] study in addition to vitamin D from the sun or supplementation and vitamin C from citrus fruits. The control group followed a habitual diet without dietary recommendation for four weeks.

The pooled effect of fibre-rich dietary intervention on the different periodontal disease outcomes assessed, compared to the habitual diet (control) is described below, and a summary is presented in Table 3.

### 3.4. Effect of Fibre on Periodontal Status-Primary Outcomes

#### Probing Depth (PD)

The study by Bartha et al. (2022) showed no significant difference within or between the intervention and control groups [30]. The Kondo et al. (2014) trial was carried out with the same study group for both high-fibre and low-fibre arms in sequential order [31]. The high-fibre intervention period in Kondo et al. (2014) showed a significant decrease in PD (*p* < 0.01) after eight weeks compared to the low-fibre control (run-in) period. Moreover, this significant reduction continued until the 24-week follow-up period [31]. There was a significant difference between intervention and control groups at three and six weeks in the Rajaram et al. (2021) [33] and at two and eight weeks in the Woelber et al. (2016) [28] studies. However, the study by Woelber et al. (2019) showed that while there were no significant differences within the intervention group (*p* = 0.904), there was an increase in PD in the control group at week 8 (*p* = 0.018) and a non-significant difference (*p* = 0.084) between intervention and control arms at week 8 [29].

The impact of the high-fibre intervention on PD was studied in four studies (*n* = 136) [28,29,30,33] included in the meta-analysis. The intervention had no statistically significant effect on PD (−0.17 mm/tooth; 95% CI, −0.37 to 0.02, *p* = 0.09) compared to controls. Significant statistical heterogeneity is indicated by an I^2^ = 77% (*p* = 0.005) (Figure 2a).

### 3.5. Clinical Attachment Loss/Level (CAL)

The Kondo et al. (2014) study recorded significantly lower CAL in both the intervention period (6.06 ± 1.39 mm, *p* < 0.05) and follow-up period (5.98 ± 1.44 mm, *p* < 0.0001) compared to the control period (6.11 ± 1.39 mm) [31]. There was a significant difference in CAL within intervention or control groups at three and six weeks in the Rajaram et al. (2021) [33] or at two and eight weeks in the Woelber et al. (2016) [28] study. CAL was not evaluated in the Bartha et al. (2022) and Woelber et al. (2019) studies [29,30].

The pooled effect of two studies (*n* = 69) [28,33] in the meta-analysis showed that intervention groups significantly reduced CAL by 0.48 mm/tooth (95% CI, −0.63 to −0.33, *p* < 0.001; I^2^ = 0%, *p* = 0.79) (Figure 2b).

### 3.6. Effect of Fibre on Periodontal Status- Secondary Outcomes

#### Bleeding on Probing (BOP)

In the study by Bartha et al. (2022), there was a significant reduction in BOP in the intervention group (*p* = 0.025) but not between groups (*p* = 0.219) [30]. Konde et al. (2014) showed a significant reduction (*p* < 0.01) in BOP% after eight weeks of the intervention (high-fibre-test-meal) period (13.2 ± 20.3% sites/tooth) compared with control (low-fibre-run-in) period (16.2 ± 22.3% sites/tooth) [31]. Here, the significant reduction (*p* < 0.05) of BOP was relatively lower in the 24-week follow-up period (14.6 ± 20.4% sites/tooth) than in the high-fibre-test-meal period.

In Rajaram et al. (2021) study, the BOP% decreased significantly in the intervention group (*p* < 0.001). It increased in the control group (*p* < 0.001) from week three to six, showing a significant difference between the groups (*p*< 0.001) [33]. The RCT by Woelber et al. (2016) showed that BOP% reduced in the intervention group (*p* < 0.001) from 53.57 ± 18.65 to 24.17 ± 11.57% while BOP% increase in the control group (*p* = 0.075) from 46.46 ± 15.61 to 64.06 ± 11.27% sites/tooth [28]. There was a significant decrease in BOP% in the intervention group (*p* < 0.001), but no differences were noted within the control group (*p* = 0.262), causing a significant difference between groups (*p* = 0.002) in Woelber et al. (2019) study [29].

Fibre-rich diet intervention significantly reduced BOP% by 27.57% sites/tooth (95% CI −50.40 to −4.74, *p* = 0.02) with significant heterogeneity (I^2^ = 94%, *p* < 0.001), which was observed from the pooled effect of four interventions (*n* = 136) [28,29,30,33] (Figure 2c).

### 3.7. Volume of Gingival Crevicular Fluid (GCF)

GCF was only measured in the SFT study by Kondo et al. (2014). It was significantly reduced (*p* < 0.01) in the intervention (high-fibre-test-meal) period (0.52 ± 0.26 µL/tooth) compared to the control (low-fibre-run-in)period (0.57 ± 0.28 µL/tooth), also significantly lower (*p* < 0.01) in the follow-up period (0.31 ± 0.25 µL/tooth) than in the intervention (high-fibre-test-meal) period [31].

### 3.8. Periodontal Inflamed Surface Area (PISA)

PISA was measured in Bartha et al. (2022), Woelber et al. (2016), and Woelber et al. (2019) studies [28,29,30]. In the Bartha et al. (2022) study, there was a significant decrease between the groups (*p* < 0.001) and in the intervention group (*p* < 0.001) but not significantly different in the control group (*p* = 0.238) [30]. In Woelber et al. (2016) study, this parameter was significantly reduced in the intervention group (*p* < 0.001) from 638.88 ± 305.41 to 284.83 ± 174.14 mm^2^ and non-significantly increased (*p* = 0.055) from 662.24 ± 420.05 to 963.24 ± 373.78 mm^2^ in the control group, with a significant difference between the groups (*p* < 0.001) [28]. No significant difference between the groups (*p* < 0.001) was observed in the Woelber et al. (2019) study [29].

Compared with controls, fibre-rich dietary intervention significantly reduced PISA by 173.88 mm^2^ (95% CI −288.06 to −59.69, *p* = 0.003) with significant high heterogeneity (I^2^ = 91%, *p* < 0.001) in the meta-analysis of three studies (n = 82) [28,29,30] (Figure 2d).

### 3.9. Plaque Index (PI)

In the Woelber et al. (2019) study, PI in both intervention (*p* = 0.0013) and control (*p* < 0.001) groups decreased significantly from week one to eight [29]. In the Bartha et al. (2022) study, PI in the intervention group was not significantly decreased (*p* = 0.274) [30]. However, the control group (*p* < 0.001) decreased significantly from week one to six, and there was a significant difference between groups (*p* < 0.001) [30]. In the Woelber et al. (2016) study, there were no significant differences between or within groups [28]. However, there was a significant difference (*p* < 0.0453) between the intervention and control groups in Rajaram et al. (2021) [33] study.

Four studies (*n* = 136) [28,29,30,33] indicated that fibre-rich dietary intervention had a significant reduction in PI by 0.02 (95% CI −0.04 to −0.00, *p* = 0.04; I^2^ = 0%, *p*= 0.98) compared to habitual diet (Figure 2e).

### 3.10. Gingival Index (GI)

In the Bartha et al. (2022) study, GI decreased significantly within the intervention group (*p* < 0.001) but not in the control group (*p* = 0.287) or between (*p* = 0.259) groups [30]. Rajaram et al. (2021) showed there was a significant decrease from 1.39 ± 0.03 to 0.80 ± 0.11 at week one to four in the intervention group (*p* < 0.001), while it decreased from 1.51 ± 0.12 to 1.49 ± 0.03 in the control group(*p* < 0.001) [33]. In the Woelber et al. (2016) study, GI decreased from 1.10 ± 0.51 to 0.54 ± 0.30 in the intervention group (*p* = 0.008), while it increased from 1.01 ± 0.14 to 1.22 ± 0.17 in the control group (*p* = 0.066) from week one to eight [28]. In Woelber et al. (2019), GI decreased significantly both between the groups (*p* = 0.042) and within the intervention (*p* < 0.001) but not significantly in the control group (*p* = 0.230) at weeks one and four [29].

Four studies (*n* = 136) [28,29,30,33] indicated that GI significantly decreased by 0.41 (95% −0.67 to −0.16, *p* = 0.002) from the pooled effect. However, significant heterogeneity was noticed (I^2^ = 73%; *p* = 0.01) (Figure 2f).

### 3.11. Oral Microbiota Outcomes

Oral microbiota outcomes were measured only in the study by Tennert et al. (2020) [32]. The median total bacterial count of the dental plaque was not significantly different between the groups [32]. The total bacteria count of saliva was also not significantly different between the groups [32]. There were no statistically significant differences in the bacterial counts of aerobic and anaerobic species in either dental plaque or saliva samples [32]. However, a significant reduction of *Streptococcus mitis* spp. (*p* = 0.025), *Granulicatella adiacens* (*p* = 0.019), *Actinomyces* spp. (*p* = 0.02), and *Fusobacterium* spp. (*p* = 0.035) in dental plaque was detected in the intervention group between the baseline and the final phase. Specific salivary species of *Actinomyces* spp. increased significantly (*p* = 0.04), while the count of the *Capnocytophaga* group was significantly reduced (*p* = 0.01) only in the control group during this time frame [32].

### 3.12. Impact of Fibre on Inflammatory Markers and Metabolic Parameters

In the Kondo et al. (2014) study, the levels of hs-CRP (high sensitivity C-reactive protein) and serum leptin at the end of the intervention (high-fibre-test-meal) period were significantly lower (*p* < 0.001) compared to the control (low-fibre-run-in) period [31]. The hs-CRP reduced from 0.55 [0.31–1.72] to 0.39 [0.19–1.36] mg/mL, and serum leptin reduced from 7.9 ± 4.5 to 4.9 ± 3.2 µg/mL. The study conducted by Kondo et al. (2014) indicated the influence of fibre intake on changes in metabolic profiles such as lipid profile, plasma glucose, and inflammatory markers [31]. Following the intervention (high-fibre-test-meal) period compared to the control (low-fibre-run-in) period, plasma glucose levels (% HbA1c) significantly decreased (from 5.4% to 5.1%, *p* < 0.01) in that study [31]. Also, compared to the control (low-fibre-run-in) period, low-density lipoprotein cholesterol (from 3.62 ± 0.70 to 3.12 ± 0.59 mmol/L, *p* < 0.001) and triglycerides (from 1.50 [1.28–2.05] to 1.11 [0.99–1.38] mmol/L, *p* < 0.01) levels significantly dropped after the intervention (high-fibre-test-meal). However, the high-density cholesterol level was not significantly different after the high-fibre-test-meal period. The study by Woelber et al. (2019) showed that inflammatory serum parameters (hs-CRP, IL-6, TNF-α) were not significantly different between or within the groups [29]. In the Bartha et al. (2022) study, there were significant (*p* = 0.007) differences between groups for body weight (BW) and body mass index (BMI) and no significant differences in waist circumference between groups [30]. Both BW and BMI were significantly reduced (*p* = 0.007) in the intervention group in the Woelber et al. (2019) study [29]. However, only BW was significantly different (*p* = 0.024) between groups. Appendix A demonstrates the baseline, follow-up values, and inter- and intra-*p* values within and between the intervention and control groups of these outcomes of the included studies.

### 3.13. Degree of Compliance to Dietary Recommendation of Fibre Intake

The studies by Rajaram et al. (2021) and Woelber et al. (2016) used regression analysis to examine the relationship between the degree of compliance with fibre intake and periodontal parameters [28,33]. Rajaram et al. (2021) showed a significant positive association between PI and fibre intake (coefficient: 0.34 ± 0.16, *p* < 0.039), but fibre intake did not show a significant association with GI (*p* = 0.970) and BOP (*p* = 0.22) [33]. Similarly, the Woelber et al. (2016) study also showed a significant positive correlation with PI and fibre (coefficient: 0.33 ± 0.15, *p* = 0.021) and no significant association with GI (*p* = 0.952), BOP (*p* = 0.11), or PISA (*p* = 0.233) [28].

### 3.14. Methodological Quality and Risk of Bias

The methodological quality assessment of six clinical trials based on the modified Jadad scale is outlined in Table 4. The study by Rajaram et al. (2021) [33] scored five out of five. Studies by Bartha et al. (2022) [30] and Woelber et al. (2019) [29] scored 4.5 out of five; Kondo et al. (2014) [31] scored one out of five; Tennert et al. (2020) [32] and Woelber et al. (2016) [28] scored three out of five.

There was no significant variation in the presence of bias within all four studies used in the meta-analysis (Figure 3 and Figure 4). Only one study demonstrated that measures have been sufficiently undertaken to adequately reduce the level of detection bias regarding the blinding of assessment outcomes [33]. The presence of bias in other areas has low risk in all studies.

## 4. Discussion

This systematic review and meta-analysis investigated for the first time if dietary fibre can be used as an intervention for managing periodontal diseases. Probing depth, CAL, BOP, PISA, PI, GI, and GCF are reflective clinical signs of periodontal inflammation in humans. The studies identified an inverse relationship between the fibre-rich daily diets and clinical periodontal markers such as PD, CAL, BOP, PISA, PI, and GI, as well as inflammatory markers and metabolic parameters in patients with periodontal diseases. Therefore, fibre-rich dietary intervention can be a practical solution for managing periodontal diseases in the early stages.

Several putative mechanisms highlighting the benefits of a high-fibre diet for oral health have been identified. Fibre might have some minor mechanical cleaning effect during chewing when they wipe over tooth surfaces. Studies have shown that fibre can inhibit oral pathogens by mechanically disturbing biofilm formation [39,40,41]. However, it is very unlikely that these described mechanical effects of the fibre have any impact on biofilm formation and bacterial colonisation in areas that really matter for the development of periodontal diseases, such as interdental areas and in the gingival sulcus or at the gingival margin. These minor effects will mainly be limited to occlusal and some of the flat buccal and lingual surfaces. Therefore, the clinical relevance of this mechanical cleaning effect of fibre can be considered at best as minor, and a little bit more effort in tooth brushing will have a greater impact on periodontal health. As high-fibre diets usually involve a greater amount of chewing, the resulting increased saliva production with the associated enzymes and antimicrobial components of the saliva may help to break down food and disturb early colonization with periodontal pathogens [42]. Moreover, vigorous chewing and increased application of masticatory forces have been shown to help maintain alveolar bone and prevent bone loss [43]. However, the clinical relevance of these effects regarding prevention of periodontal diseases has to be considered as minor.

The inflammatory process of periodontal diseases produces reactive oxidative species (ROS) as part of the regular defense response to plaque biofilm, exacerbating periodontal breakdown [44]. During the degradation of dietary fibre, a series of antioxidant and anti-inflammatory compounds are released into the host’s bloodstream [45]. Antioxidants may improve periodontal health and outcomes of periodontal therapy by reducing oxidative stress via scavenging ROS [46]. Kim et al. (2012) found that β-glucan (a form of dietary fibre) has an antioxidant effect and effectively inhibits periodontitis and related alveolar bone loss [47]. This study also found that β-glucan fibre lowered the periodontal inflammatory markers such as myeloperoxidase, interleukin-1beta (IL-1β), and tumor necrosis factor-alpha (TNF-α) in gingival tissue [47].

In the gut, high-fibre diets can significantly alter the intestinal environment by influencing the gut microbiome, gastrointestinal immune and endocrine responses, the nitrogen cycle, and microbial metabolism [48]. Consequently, changes in gut physiology and biochemistry can affect other major organs involved in nutrient management and detoxification, which also improve outcomes of periodontal therapy [49].

Haematogenous dissemination of periodontal bacteria or transfer of inflammatory mediators from leaky and inflamed periodontal tissues to the bloodstream results in systemic inflammation [50]. They can activate and stimulate white blood cells in the bloodstream [51], putting them on high alert and releasing damaging mediators such as oxygen radicals and cytokines, which drive inflammation [52]. This inflammation then causes systemic consequences by allowing oral bacteria to spread throughout the body and promotes the development of systemic diseases directly or indirectly by producing endotoxins [53,54].

However, as dietary fibre has a protective effect on periodontal inflammation [22,55], severity can be mitigated. For instance, inflammatory markers such as highly sensitive C-reactive protein (hs-CRP) and plasma tissue plasminogen activator-1 (tPAI-1) can be reduced by having a high-fibre diet [31,56]. The breakdown of dietary fibre by the colonic microbiota has a positive effect in preventing systemic inflammation and conditions such as diabetes [57], cardiovascular disease [58], and obesity [59]. A high-fibre diet may also reduce lipid oxidation, which, in turn, may reduce systemic inflammation [56]. As systemic inflammation and the range of the above-mentioned diseases exacerbate the periodontal disease condition, a high-fibre diet may positively affect periodontitis and also reduce the systemic inflammatory burden. Other positive effects of consuming a high-fibre diet also include maintaining beneficial bacteria in the gut, which secrete chemicals that reduce inflammation throughout the body [60,61]. In addition, fermentation of dietary fibre such as oligosaccharide, polysaccharide, and resistant starch by colonic anaerobic microbes produces short-chain fatty acids (SCFAs) such as acetate, butyrate, and propionate [62], which play essential roles in regulating host metabolism and immunity [62]. High-fibre diet is linked to reducing plasma levels of hs-CRP, IL-6, and TNF-R2 by facilitating the anti-inflammatory process [63,64]. In this anti-inflammatory process, dietary fibre decreases glucose and lipid oxidation while ensuring a healthy intestinal environment [65]. Also, dietary fibre may reduce inflammation by modifying adipokines in adipose tissue and increasing lipid and lipophilic compound circulation in the intestine [66]. Dietary fibre is also well known for lowering plasma glucose excursions following meals high in fibre [67]. Moreover, fibre has been recognised as one positive factor in reducing the risk of all forms of systemic diseases, as fibre-rich diets have antioxidants and lower levels of saturated fat [68]. Furthermore, dietary fibre byproducts such as glucans and mannan-oligosaccharides reduce the accumulation of pro-inflammatory cytokines IL-1 and TNF-α in local periodontal tissue [69,70,71]. Previous studies on diabetic patients have shown that high-fibre intake can help diabetic patients maintain better metabolic control with reduced levels of systemic inflammatory markers and improved periodontal indices [69,70,71].

The effect of fibre consumption on inflammatory markers together with metabolic parameters was only studied by Bartha et al. (2022) [30], Kondo et al. (2014) [31], and Woelber et al. (2019) [29] in this review, despite the link relating inflammation and metabolic effect to periodontal diseases [72]. Also, given that they observed a difference in inflammatory markers after a high-fibre diet intervention, some of the positive effects on periodontal indices observed in these studies may be due to a decrease in systemic inflammation. Inflammatory markers, such as hs-CRP, IL-6, and TNF-α-R2 levels in the blood, were negatively correlated with dietary fibre intake [73] by reducing body weight or limiting obesity-associated systemic inflammation [19,20]. As a higher BMI is a risk factor for metabolic disorders and systemic inflammation, a sustained pattern of a high-fibre diet, which has lower energy density, is linked with a lower BMI and an anti-inflammatory profile [74,75].

Furthermore, it is essential to consider the impact of antibiotics on periodontal health in this context, as it opens avenues to explore the potential role of dietary fibre as a prebiotic intervention. While antibiotics are a common approach to combat oral infections, including prophylactic use during intraoral procedures, their broad-spectrum application can significantly affect periodontal disease development [76]. However, the integration of dietary fibre into periodontal care presents an opportunity to address these concerns as it’s known for selectively nourishing beneficial gut bacteria, which may also exert positive effects on the oral microbiome [9]. By fostering the growth of probiotic strains that support oral health, dietary fibre intake could potentially mitigate the reliance on antibiotics. This impact is significant even when antibiotics are prescribed directly for periodontitis management. Notably, the impact of antibiotics on periodontal health remains substantial even when prescribed specifically for periodontal treatment. Of particular concern is the development of antibiotic resistance among periodontal microorganisms, posing a serious challenge in the treatment of periodontal illnesses [9].

This review has identified several limitations that need to be acknowledged. The dietary interventions analyzed in the included studies exhibited variations in study design, participant health status, intervention duration, and methods of dietary assessment. These disparities collectively hinder the establishment of definitive conclusions. Furthermore, it is essential to recognise that the participant pool within the selected studies primarily consisted of gingivitis, a reversible and early-stage form of periodontitis. The enrollment of such participants, coupled with their active engagement, could have led to an increased effort in oral hygiene practices, including tooth brushing. Consequently, noticeable impacts on various assessed periodontal parameters, particularly in the context of gingivitis, may have ensued. In this light, the observed changes in parameters such as PD and CAL warrant caution interpretation. It is crucial to note that, by definition and as per the classification of periodontal diseases, individuals with gingivitis do not exhibit a PD of ≥3 mm, which is categorised as physiological PD. Moreover, they do not display any attachment loss. Consequently, alternation in the clinical parameters such as PD and CAL primarily reflects shifts in gingival inflammation or soft tissue conditions rather than substantial changes in periodontal health. Additionally, the employment of diverse methods for measuring clinical outcomes across the studies introduced a degree of variability in the recorded values. For instance, PD and CAL measurements were conducted using a conventional manual periodontal probe in all four studies. However, it is important to note that the magnitude of changes in CAL lacks clinical significance, and manual probing might not be sufficiently sensitive to detect differences of such a minute scale. The unitisation of a meticulously calibrated Florida Probe would offer a more precise approach to PD and CAL measurements. While the objective of this review was to encompass a spectrum of periodontal diseases, the reality remains that the subjects featured in the included studies only represented individuals with gingivitis. Furthermore, it is worth acknowledging that these studies also accounted for various dietary constituents, including omega-3/6, Vitamin D, Vitamin C, and antioxidants. Consequently, certain observed effects could potentially be attributed to the presence of these nutrients within the diet. This intricate interplay of multiple factors underscores the need for more comprehensive research to unravel the distinct contributions of dietary fibre in the context of periodontal diseases.

The relationship between fibre and periodontal disease outcomes in a mixed diet remains unknown. Although one study demonstrated the compliance of various nutrients, possibly linked to periodontal disease markers, with clinical outcomes, this aspect remains unexplored. All the studies lacked information regarding subjects’ routines for oral hygiene practices, gut health, bowel movements, or physical activity, all of which could have influenced the interpretation of results. Additionally, there was a need to differentiate between various sources of dietary fibre. As a result, these factors could potentially limit the conclusions drawn from this systematic review, and prudence is required when making generalisations.

However, to the best of the authors’ knowledge, this systematic review and meta-analysis represents the first attempt to gather high-quality evidence from intervention studies, considered the highest level of research design, to ascertain whether a high-fibre diet exerts positive effects on periodontal disease markers. Notably, all studies employed in this review were conducted as pragmatic trials, implying that the findings are directly applicable to real-world situations rather than being based on supplementary or animal studies. Three out of four interventions across the studies followed similar dietary recommendations. Nonetheless, there remains a need for further RCTs involving patient populations with both active and a history of periodontitis to establish a more definitive link. This will be essential for future systematic reviews seeking to provide more conclusive evidence.

As far as dietary recommendations are concerned, high-fibre foods, such as whole grains [77], fruits, and vegetables [21,78], can be recommended as a means to mitigate the risk of periodontal diseases, slow its progression, and enhance the healing process. Furthermore, other natural substances such as probiotics (e.g., *Lactobaccilus*, *Bifidobacterium*, *Streptococcus*, and *Weissella*) [79], paraprobiotic (inactivated microbial cells), and postbiotics (substances released through the metabolic activity of the microorganism without containing the viable microorganisms themselves) offer economical and natural avenues for combating periodontal diseases and should be considered as future goals [80].

## 5. Conclusions

In summary, the meta-analysis findings indicate that a dietary intervention rich in fiber led to a significant reduction in various periodontal parameters, including CAL, BOP, PISA, PI, and GI, while also showing a positive but statistically non-significant trend in PD. These results suggest that the incorporation of a fiber-rich daily diet may offer promise as a complementary approach for the prevention and management of periodontal diseases. This presents a valuable public health message for individuals dealing with periodontal issues. Furthermore, it is worth noting that food education can be a proactive measure when combined with home dental hygiene education. This holistic approach to oral health can potentially yield more comprehensive and sustainable benefits for individual seeking to improve their periodontal well-being.

## Figures and Tables

**Figure 1 nutrients-15-04034-f001:**
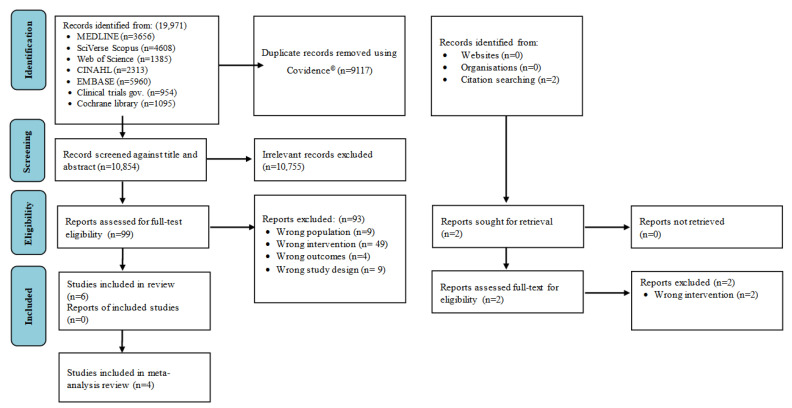
Study screening and identification flow chart.

**Figure 2 nutrients-15-04034-f002:**
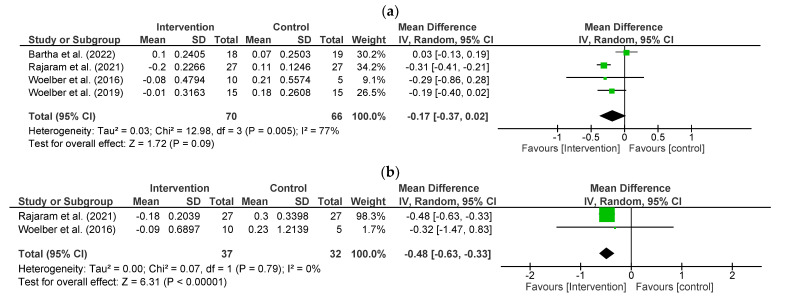
Effect of fibre-rich diet on: (**a**) Probing Depth (mm/tooth) [28,29,30,33]; (**b**) Clinical Attachment Level/Loss (mm/tooth) [28,33]; (**c**) Bleeding On Probing (% sites/tooth) [28,29,30,33]; (**d**) Periodontal Inflamed Surface Area (mm^2^) [28,29,30]; (**e**) Plaque Index [28,29,30,33]; (**f**) Gingival Index [28,29,30,33]. The green squares represent each studies individual standardised mean difference (SMD) and the extending lines the confidence intervals. The black diamond is a visual representation of the pooled SMD and its confidence intervals.

**Figure 3 nutrients-15-04034-f003:**
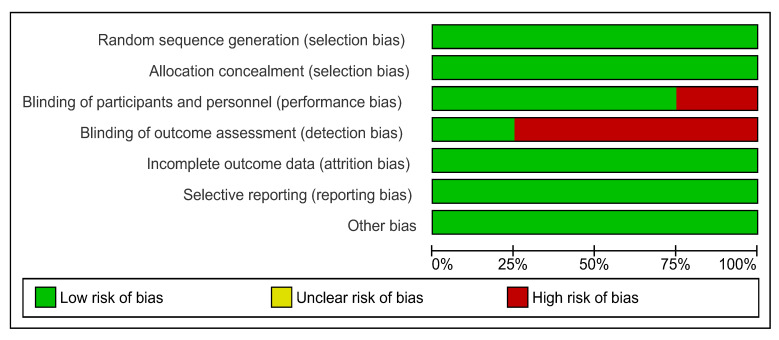
Risk of bias graph (Cochrane risk of bias tool): review authors’ judgments about each risk of bias item presented as percentages across all included studies. RevMan Version 5.3. Copenhagen: The Nordic Cochrane Centre, the Cochrane Collaboration, 2014.

**Figure 4 nutrients-15-04034-f004:**
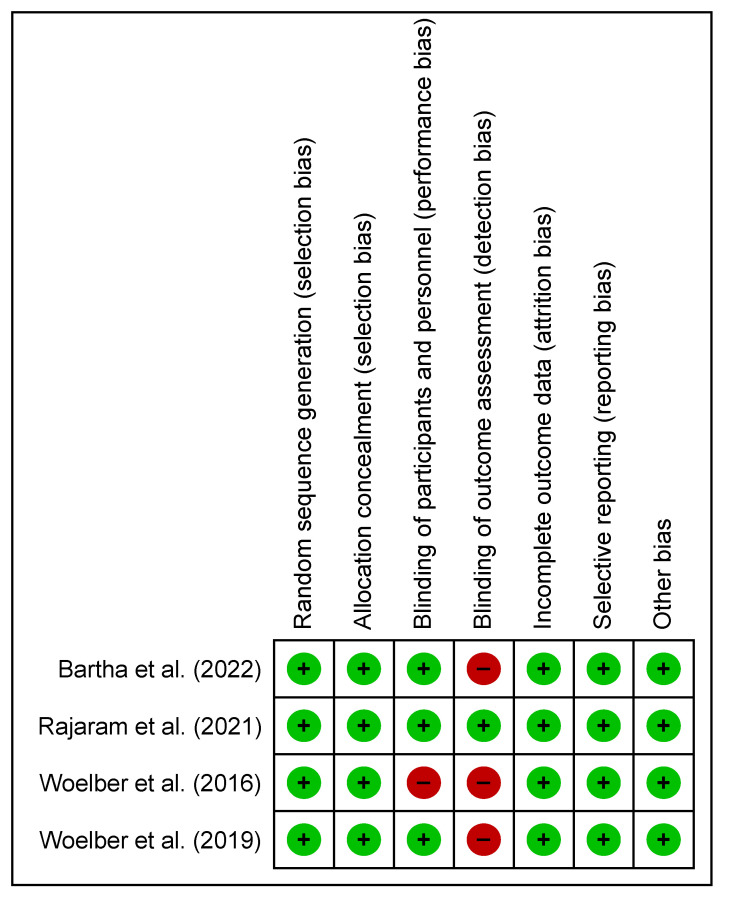
Risk of bias summary (Cochrane risk of bias tool) for included studies: review authors’ judgments about each risk of bias item for each included study [28,29,30,33]. Green circle with plus sign indicates low risk of bias and red circle with minus sign indicates high risk of bias. RevMan Version 5.3. Copenhagen: The Nordic Cochrane Centre, the Cochrane Collaboration, 2014.

**Table 1 nutrients-15-04034-t001:** Human interventional studies conducted investigating the effect of dietary fibre on periodontal diseases primary outcomes.

Study Details	Study Design; Duration	Diagnosis Criteria Defining the Periodontal Diseases	Study Population; Sample Size (I:C); Mean Age (I:C)	Form of Giving of Dietary Fiber; Dietary Fiber Intake/Day (g)	Outcomes Evaluated
Bartha et al. Germany; (2022) [30]	RCT, SB; 6 weeks	Generalized gingivitis (BOP > 30%)	Patients with gingivitis; 37 (18:19); 32.71 ± 8.87:29.21 ± 7.17 years	Mediterranean diet including fibre rich foods; 22.10–26.06 g	*Periodontal outcomes:* PD, GI, BOP, PI, PISA *Physical outcomes:* BW, BMI, WC
Kondo et al. Japan; (2014) [31]	SFT; 8 weeks	No specific periodontal diseases criteria mention.	High-risk subjects (BMI of at least 25.0 kg/m^2^ or impaired glucose tolerance); 17 (17:17); 45.0 ± 6.5:45.0 ± 6.5 years	Recommended diet including fibre rich foods; 30.3 g	*Periodontal outcomes:* PD, CAL, BOP, GCF *Inflammatory markers:* hs-CRP *Physicaloutcomes:* BMI, WC *Other serum parameters:* Lipid profile, plasma glucose, serum leptin
Rajaram et al. India; (2021) [33]	RCT, DB; 4 weeks	Gingivitis (GI > 0.5 and ≤3)	Patients with gingivitis; 54 (27:27); 36.1 ± 8.3:35.2 ± 7.2 years	Recommended diet including fibre rich foods(from fruits and vegetables); Amount not specified	*Periodontal outcomes:* PD, CAL, PI, GI, BOP%
Woelber et al. Germany; (2016) [28]	RCT; 8 weeks	Mild gingivitis (GI = 1.10)	Patients with gingivitis; 15 (10/5); 34.4 ± 14.1:34.0 ± 16.5 years	Recommended diet including fibre rich foods(from fruits and vegetables); Amount not specified	*Periodontal outcomes:* PD, CAL, BOP, PISA, PI, GI
Woelber et al. Germany; (2019) [29]	RCT, SB; 8 weeks	Gingivitis (GI ≥ 0.5)	Patients with gingivitis; 30 (15:15); 27.2 ± 4.7:33.7 ± 13.1 years	Recommended diet including fibre rich foods (from vegetables, fruits, legumes, bran); Amount not specified	*Periodontal outcomes:* PD, PI, GI, BOP, PISA *Inflammatory markers:* hs-CRP, IL-6, TNF-α *Physicaloutcomes:* BW, BMI, WC
Tennert et al. Switzerland; (2020) [32]	RCT; 4 weeks	Gingivitis (GI > 0.5)	Patients with gingivitis; 14 (9:5); 34.0 (24–63) years	Oral health optimized diet including fibre rich foods (fruits and vegetables) Amount not specified	Only Oral microbiota outcomes

BOP: Bleeding On Probing; BMI: Body Mass Index; BW: Body Weight; C: Control group; CAL: Clinical Attachment Level/Loss; DB: Double blind; GCF: Gingival Crevicular Fluid; GI: Gingival Index; hs-CRP: high-sensititve C-reactive protein; I: Intervention group; IL-6: Interleukin 6; PD: Probing (pocket) Depth; PI: Plaque Index; PISA: Periodontal Inflamed Surface Area; RCT: Randomised Controlled Trial; SB: Single blind; SFT: Sequential Feeding Trial (non-randomized); TNF: Tumor Necrosis Factor; WC: Waist Circumference.

**Table 2 nutrients-15-04034-t002:** Periodontal outcomes of included studies.

Parameter		Bartha et al. (2022) [30]			Kondo et al. (2014) [31]			Rajaram et al. (2021) [33]			Woelber et al.(2016) [28]			Woelber et al. (2019) [29]		
		Intervention (I) Mean ± SD	Control (C) Mean ± SD	Inter-*p* (I vs. C)	Intervention (I) Mean ± SD	Control (C) Mean ± SD	Inter-P (I vs. C)	Intervention (I) Mean ± SD	Control (C) Mean ± SD	Inter-P (I vs. C)	Intervention (I) Mean ± SD	Control (C) Mean ± SD	Inter-P (I vs. C)	Intervention (I) Mean ± SD	Control (C) Mean ± SD	Inter-P (I vs. C)
		Mean ±SD														
**PD** **(mm/tooth)**	Baseline (BL)	2.26 ± 0.18	2.29 ± 0.18	0.616	2.28 ± 0.74	-	-	2.43 ± 0.07	2.22 ± 0.01	**<0.001**	2.19 ± 0.34	2.31 ± 0.43	0.564	1.85 ± 0.27	1.82 ± 0.24	0.750
	Follow-up (FU)	2.36 ± 0.17	2.36 ± 0.18	1.000	2.21 ± 0.77	-	-	2.23 ± 0.01	2.33 ± 0.03	**<0.001**	2.11 ± 0.35	2.52 ± 0.40	0.062	1.84 ± 0.17	2.00 ± 0.14	**0.009**
	Intra-P (BL vs. FU)	0.096	0.239		0.789	-		**<0.001**	**<0.001**		0.611	0.447		0.940	0.018	
**CAL** **(mm)**	Baseline (BL)	-	-	-	6.11 ± 1.39	-	-	2.35 ± 0.06	2.49 ± 0.10	**<0.001**	2.31± 0.52	2.53 ± 0.90	0.554	-	-	-
	Follow-up (FU)	-	-	-	6.06 ± 1.39	-	-	2.17 ± 0.05	2.79 ± 0.13	**<0.001**	2.22 ± 0.47	2.76 ± 0.88	0.139	-	-	-
	Intra-P (BL vs. FU)	-	-		0.9171	-		**<0.001**	**<0.001**		0.670	0.694		-	-	
**%BOP** **(% sites/tooth)**	Baseline (BL)	51.00 ± 14.65	43.21 ± 14.25	0.110	16.20 ± 22.3	-	-	53.88 ± 1.75	43.62 ± 2.49	**<0.001**	53.57 ± 18.65	46.46 ± 15.61	0.478	30.35 ± 11.07	28.39 ± 13.32	0.665
	Follow-up (FU)	39.93 ± 13.74	39.74 ± 11.0	0.963	13.20 ± 20.3	-	-	23.55 ± 1.79	68.34 ± 0.88	**<0.001**	24.17 ± 11.57	64.06 ± 11.27	**<0.001**	23.55 ± 13.61	27.09 ± 10.03	0.424
	Intra-P (BL vs. FU)	**0.025**	0.406		0.684	-		**<0.001**	**<0.001**		**<0.001**	0.075		0.145	0.765	
**GCF** **(** **µL/tooth)**	Baseline (BL)	-	-	-	0.57 ± 0.28	-	-	-	-		-	-	-	-	-	-
	Follow-up (FU)	-	-	-	0.52 ± 0.26	-	-	-	-		-	-	-	-	-	-
	Intra-P (BL vs. FU)	-	-		0.593	-		-	-		-	-		-	-	
**PISA** **(mm^2^)**	Baseline (BL)	616.33 ± 201.39	528.94 ± 173.48	0.165	-	-	-	-	-	-	638.88 ± 305.41	662.24 ± 420.05	0.903	315.20 ± 148.68	270.50 ± 140.97	0.405
	Follow-up (FU)	512.02 ± 205.83	514.26 ±148.79	0.970	-	-	-	-	-	-	284.83 ± 174.14	963.24 ± 373.78	**<0.001**	252.37 ± 151.78	286.00 ± 114.02	0.498
	Intra-P (BL vs. FU)	0.134	0.781		-	-		-	-		**0.005**	0.266		0.261	0.273	
**PI**	Baseline (BL)	1.51 ± 0.21	1.37 ± 0.38	0.178	-	-	-	0.84 ± 0.01	0.87 ± 0.01	**<0.001**	0.77± 0.52	0.75± 0.63	0.949	0.56 ± 0.27	0.57 ± 0.19	0.908
	Follow-up (FU)	1.49 ± 0.24	1.39 ± 0.24	**<0.001**	-	-	-	0.86 ± 0.02	0.91 ± 0.01	**<0.001**	0.84 ± 0.47	0.97 ± 0.70	0.674	0.48 ± 0.13	0.48 ± 0.12	1.000
	Intra-P (BL vs. FU)	0.792	**<0.001**		-	-		**<0.001**	**<0.001**		0.756	0.616		0.310	0.132	
**GI**	Baseline (BL)	1.30 ± 0.25	1.11 ± 0.42	0.143	-	-	-	1.39 ± 0.03	1.51 ± 0.12	**<0.001**	1.10 ± 0.51	1.01 ± 0.14	0.709	0.92 ± 0.14	0.83 ± 0.22	0.192
	Follow-up (FU)	0.99 ± 0.22	0.97 ± 0.27	0.826	-	-	-	0.80 ± 0.11	1.49 ± 0.03	**<0.001**	0.54 ± 0.30	1.22 ± 0.17	**<0.001**	0.61 ± 0.29	0.74 ± 0.18	0.151
	Intra-P (BL vs. FU)	**<0.001**	0.286		-	-		**<0.001**	0.405		0.008	0.066		**<0.001**	0.230	

BL: Baseline; BOP: Bleeding On Probing; C: Control group; CAL: Clinical Attachment Level/Loss; FU: Follow-up; GCF: Gingival Crevicular Fluid; GI: Gingival Index; I: Intervention group; PD: Probing (pocket) Depth; PI: Plaque Index; PISA: Periodontal Inflamed Surface Area. The numbers marked in bold indicate they are statistically significant.

**Table 3 nutrients-15-04034-t003:** Pooled estimates of fibre-rich dietary intervention on outcomes evaluated in meta-analysis (summary of forest plots).

Periodontal Outcome (Unit)			All Trials *		
	N	n	Pooled Estimate (95% CI)	*p* Value (Pooled Estimate)	I^2^ (%), *p* Value of I^2^
Probing Depth (mm/tooth)	4	136	−0.17 (−0.37, 0.02)	0.09	77, 0.005
Clinical Attachment Loss (mm/tooth)	2	69	−0.48 (−0.63, −0.33)	<0.001	0, 0.79
Bleeding on Probingm (% sites/tooth)	4	136	−27.57(−50.40, −4.74)	0.02	94, <0.001
Periodontal Inflamed Surface Area (mm^2^)	3	82	−173.88 (−288.06, −59.69)	0.003	91, <0.001
Plaque Index	4	136	−0.02 (−0.04, −0.00)	0.04	0, 0.98
Gingival Index	4	136	−0.41 (−0.67, −0.16)	0.002	73, 0.01

* Pooled estimates (pooled mean difference in above parameters between intervention and control) were calculated based on all eligible trials included in the meta-analysis. N = number of studies, n= number of participants, I^2^ = heterogeneity.

**Table 4 nutrients-15-04034-t004:** Reporting methodological quality of the included studies assessed by modified Jadad scale.

Characteristics of Modified Jadad Scale	Bartha et al. (2022) [30]	Kondo et al. (2014) [31]	Rajaram et al. (2021) [33]	Woelber et al. (2016) [28]	Woelber et al. (2019) [29]	Tennert et al. (2020) [32]
Was the study characterized as being random? (1 or 0)	1	0	1	1	1	1
Was the study referred to as being double-blind? (1 if double-blind, 0.5 if single blind or 0 if no-blinded)	0.5	0	1	0	0.5	0
Was a description given of dropouts and withdrawals? (1 or 0)	1	1	1	1	1	1
Was the randomization technique outlined in the paper appropriate? (1 or 0)	1	0	1	1	1	1
Was the described and appropriate blinding technique used? (1 or 0)	1	0	1	0	1	0
Was the paper’s description of the randomization method inappropriate? (0 or −1)	0	0	0	0	0	0
Was the described blinding technique inappropriate? (0 or −1)	0	0	0	0	0	0
**Total**	**4.5**	**1**	**5**	**3**	**4.5**	**3**

## Data Availability

No new data were created or analyzed in this study. Data sharing is not applicable to this article.

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
