# Peer review of "Role of Dietary Fibre in Managing Periodontal Diseases—A Systematic Review and Meta-Analysis of Human Intervention Studies"

_nutrients, 2023, doi:10.3390/nu15184034_

Round 1
Reviewer 1 Report
Manuscript of considerable interest for the dental sector, it needs a major revision before evaluating its publication.
Abstract: the introductory part is very copious, reduce it and emphasize the results obtained more.
Keywords: very generic, add specific ones and consider registering them on MeSH.
Introduction: the description of the new classification of periodontal disease is missing, and above all how the microbiota changes in the periodontal patient compared to the implant one (Butera et al.) and all the minimally invasive systems to maintain a balanced microbiota (Scribante et al. Microorganisms mdpi) .
Ok that the introduction of fibers reduce clinical indices but if we have large risk factors they wouldn't be enough.
Materials methods: too many selected search engines, limit yourself to scopus Web of science, the others are not indexed journals. Review the calculation.
Results, very confusing, reorganize the tables to make them more usable to the reader.
Discussion: add the type of foods to recommend to patients, it is not enough to indicate only fibers and add all natural substances as future goals. (probiotics, paraprobiotics, postbiotics).
Conclusion: add food education as a proactive action combined with home oral hygiene education.
Add references required.
Author Response
Answers for 1st Reviewer’s comments
General Comment: Manuscript of considerable interest for the dental sector, it needs a major revision before evaluating its publication.
Response: Thank you very much for taking the time to review this manuscript. Please find the responses below and the corresponding revisions highlighted in the re-submitted files.
Comments 1: Abstract: the introductory part is very copious, reduce it and emphasize the results obtained more.
Response: Thank you for your feedback. We have revised the abstract to be more concise, with a greater emphasis on highlighting the obtained results.
Comments 2: Keywords: very generic, add specific ones and consider registering them on MeSH.
Response: Thank you for your suggestion. Keywords already presented were generated using MeSH keywords from the selected PubMed articles which is considered as one of the three main methods available to search for said keywords. However, we now searched the keywords using the MeSH on Demand tool (https://meshb.nlm.nih.gov /MeSHonDemand) and updated the MeSH keywords. (Line 32-33)
Comments 3: Introduction: the description of the new classification of periodontal disease is missing, and above all how the microbiota changes in the periodontal patient compared to the implant one (Butera et al.) and all the minimally invasive systems to maintain a balanced microbiota (Scribante et al. Microorganisms mdpi) .
Ok that the introduction of fibers reduce clinical indices but if we have large risk factors they wouldn't be enough.
Response: We thank the reviewers for their valuable suggestion, we have now added a section on the role of diet on periimplant microbiota (line 57-64), new classification system (67-69) and important of minimally invasive procedure like dietary intervention for managing periodontal disease (line 83-85).
Comments 4: Materials methods: too many selected search engines, limit yourself to Scopus, Web of science, the others are not indexed journals. Review the calculation.
Response: Thank you for your observation. We appreciate your concern regarding the number of selected search engines. Our choice to include multiple search engines in our systematic review was influenced by established guidelines for systematic reviews. These guidelines typically recommend the use of at least three databases to ensure a comprehensive search for relevant studies.
Initially, we conducted searches using Scopes, Web of Science, and PubMed, which are indexed and widely recognised databases. However, our initial searches yielded a limited number of papers relevant to our systematic review.
To meet the rigorous standards of systematic review requirements and avoid potential gaps in our search, we opted to include additional sources, such as Cochrane, which is considered a gold standard for clinical systematic reviews due to its comprehensive coverage of high-quality, independent evidence. We also included Embase, a database known for covering crucial international biomedical literature, as well as CINHL, which provides access to citations and abstracts of scholarly, peer-reviewed articles across various publications. Furthermore, the inclusion of ClinicalTrials.gov allowed us to identify ongoing or completed clinical trials related to our topic.
Our rationale for utilising multiple search engines was driven by the principle that a robust search strategy aims to achieve a high recall of relevant references within a single database while also enhancing search accuracy. This approach reduces the risk of overlooking pertinent papers. By employing seven search engines, we aimed to ensure confidence in capturing nearly all relevant studies pertinent to our systematic review.
Comments 5: Results, very confusing, reorganize the tables to make them more usable to the reader.
Response: Thank you for your feedback. We have reorganized all the tables to make them more reader-friendly and less confusing in the Results section.
Comments 6: Discussion: add the type of foods to recommend to patients, it is not enough to indicate only fibers and add all natural substances as future goals. (probiotics, paraprobiotics, postbiotics).
Response: Thank you for your suggestion. We have added an additional paragraph (lines 601-607) in the Discussion section to address the type of foods to recommend to patients, including the mention of fibers and future goals involving natural substances like probiotics, paraprobiotics, and postbiotics.
Comments 7: Conclusion: add food education as a proactive action combined with home oral hygiene education.
Response: Thank you for your suggestion. We have incorporated your recommendation by adding a point about food education as a proactive action combined with home oral hygiene education in the Conclusion (lines 615-616).
Comments 8: Add references required.
Response: Thank you for your comment. We appreciate your feedback. We have included the required references for the new introduction and discussion sentences

Reviewer 2 Report
row 103: space between "areca" and "nut"
row new 48: "compliance was0.50, 0.64, 0" space before the first number
row new 79: "at the 24week follow-up period" insert space after number
Congratulation for a well conducted and writen meta-analysis
Author Response
Thank you very much for taking the time to review this manuscript. Please find the responses below and the corresponding revisions highlighted in the re-submitted files.
Comments 1: Row 103: space between "areca" and "nut"
Response: Thank you for bringing this to our attention. Now it's been corrected
Comments 2: Row new 48: "compliance was 0.50, 0.64, 0" space before the first number
Response: Thank you for pointing this out. Space was given now.
Comments 3: row new 79: "at the 24week follow-up period" insert space after number
Response: Thank you. Space was given after the number.

Reviewer 3 Report
Please see attached PDF for specific comments

English is fine
Author Response
Answers for 3rd Reviewer’s comments
Thank you very much for taking the time to review this manuscript. Please find the responses below and the corresponding revisions highlighted in the re-submitted files.
Comments 1: The abstract is too long, please shorten it.
Response: Thank you for your suggestion. The abstract has now been shortened.
Comments 2: A flow diagram of the methodology would be useful in visualizing the process.
Response: Authors thank the reviewer for the suggestion. PRISMA flow diagram (Figure 1) already visually represents the methodology process. Therefore, we suggest not introducing a new flow diagram for this systematic review process. However, we are open to considering it if the reviewer still finds it necessary.
Comments 3: The authors should also mention the influence of antibiotics in the development of periodontal disease, not necessarily taken specifically for the treatment of periodontitis.
I suggest:
Martu, I.; Goriuc, A.; Martu, M.A.; Vata, I.; Baciu, R.; Mocanu, R.; Surdu, A.E.; Popa, C.; Luchian, I. Identification of Bacteria Involved in Periodontal Disease Using Molecular Biology Techniques. Rev. Chim. 2017, 68, 2407–2412.
Response: We appreciate the reviewer's suggestion. It's crucial to recognize that antibiotics can impact the development of periodontal disease, even when they are not prescribed specifically for periodontitis treatment. Antibiotic resistance among periodontal microorganisms has become a significant concern in the context of treating periodontal diseases. We have now incorporated this point, along with the provided reference, into our revised manuscript (Page 19, lines 178-185).
Comments 4: The conclusions section are too long, please transfer and reformulate the last 2 sentences at the end of the previous sections.
Response: Thank you for your suggestion. Conclusion section was shortened and revised.

Round 2
Reviewer 1 Report
The manuscript has been revised successfully, it can be published
Reviewer 3 Report
The manuscript has been improved